# “Outsourcing” Diatoms in Fabrication of Metal-Doped 3D Biosilica

**DOI:** 10.3390/ma13112576

**Published:** 2020-06-05

**Authors:** Weronika Brzozowska, Myroslav Sprynskyy, Izabela Wojtczak, Przemysław Dąbek, Andrzej Witkowski, Bogusław Buszewski

**Affiliations:** 1Institute of Marine and Environmental Sciences, University of Szczecin, Mickiewicza 16, 70-383 Szczecin, Poland; weronika.brzozowska@phd.usz.edu.pl (W.B.); pdabek@usz.edu.pl (P.D.); Andrzej.Witkowski@usz.edu.pl (A.W.); 2Department of Environmental Chemistry and Bioanalytics, Faculty of Chemistry, Nicolaus Copernicus University in Toruń, 7 Gagarina Str., 87-100 Toruń, Poland; izabelawojtczak1991@gmail.com (I.W.); bbusz@umk.pl (B.B.); 3Centre for Modern Interdisciplinary Technologies, Nicolaus Copernicus University, Wilenska 4, 87-100 Torun, Poland

**Keywords:** diatoms, diatomaceous biosilica, metal doping, metabolic inserting

## Abstract

Diatoms have an ability that is unique among the unicellular photoautotrophic organisms to synthesize an intricately ornamented siliceous (biosilica) exoskeleton with an ordered, hierarchical, three-dimensional structure on a micro- to nanoscale. The unique morphological, structural, mechanical, transport, photonic, and optoelectronic properties of diatomaceous biosilica make it a desirable material for modern technologies. This review presents a summary and discussion of published research on the metabolic insertion of chemical elements with specific functional activity into diatomaceous biosilica. Included in the review is research on innovation in methods of synthesis of a new generation of functional siliceous materials, where the synthesis process is “outsourced” to intelligent microorganisms, referred to here as microtechnologists, by providing them with appropriate conditions and reagents.

## 1. Introduction

The use of microorganisms, especially unicellular microalgae, is a novel approach in the design and synthesis of new inorganic composite nanomaterials [1,2]. Some microorganisms have the ability to synthesize unique mineral composites with complex, hierarchical structures on a micro- to nanoscale [3]. The single-celled photoautotrophic microorganisms—diatoms (Bacillariophyceae)—have an astonishing variety of intricately ornamented siliceous exoskeletons, called frustules or valves, with a unique three-dimensional structure (Figure 1 [4]) in more than 100,000 known species [5]. Such a variety of unique, precise siliceous structures with orderly pore (areola) systems makes them a desirable material for modern technologies [6,7,8,9,10,11,12,13,14].

The concept of using diatomaceous biosilica as an implementation material in modern technologies, especially in nanotechnology, is relative new and was proposed in 1994 by Gordon R. and Drum R. W. [15]. Since then, the phenomenal ability of diatoms to synthesize unique three-dimensional structures with specific physicochemical (optical, electrical, filtration, thermal, mechanical) properties from amorphous silica has held growing fascination for biologists, chemists, and physicists [6,16,17,18,19,20,21]. Currently, diatomaceous biosilica, due to its three-dimensional, porous structure, wide availability, and the possibility of biosynthesis through the cultivation of diatoms under artificial conditions, is one of the most frequently used substitutes for mesoporous silica materials in modern technologies. These materials, despite their biocompatibility and large specific surface area [22], are difficult to synthesize because of the necessity of considerable financial input, a large amount of energy, and an association of toxic materials using [23].

The unique, hierarchically porous 3D structure of diatom frustules makes them an attractive source of solutions for the development of modern material engineering. There are a wide range of possibilities for the use of such materials, e.g., in the production of biosensors, optical devices, catalysts, semiconductors, effective adsorbents, templates for nanolithography, and in designing drug carriers or bone implants [6,11,17,18,20,23,24,25]. The range of perspectives for the use of diatomaceous biosilica is shown in Figure 2.

Diatomaceous biosilica can be successfully used as an electrode material for energy generation and storage, or as photonic crystals [26,27,28,29,30,31]. Diatom frustules can be used as microlenses, as they are able to focus light below the diffraction limit, and their ability to accumulate high-intensity light can lead to the development of modern solar cells [32,33,34,35,36]. High thermal and mechanical resistance as well as unique optical properties make diatomaceous biosilica an inspiration in the development of modern optoelectronic devices [23,24,30,37]. However, many of the possible applications for diatomaceous biosilica as an industrial material are limited by the chemistry of the silica in diatom frustules. For this reason, considerable efforts have been made recently to modify the structure of diatom frustules to make them more technologically functional, whilst preserving their unique shape and morphology [23,33,38,39,40,41,42]. An extremely exciting proposal for the modification of biosilica is its reduction to pure silicon, without destroying its three-dimensional structure, which would be associated with new, broad possibilities in the field of microelectronics [18,38]. Promising results have been obtained using diatomaceous biosilica as a matrix in the chemical synthesis of nanomaterials [20,27,43,44]. Umernura et al. [41] proposed using fragmented diatomaceous biosilica as a matrix for luminescence in the liquid phase. The potential for placing specific proteins, enzymes, or antibodies within the structure of diatoms could allow for the production of microchip-sized hybrid biosensors, which would be a medical breakthrough [16,18,45].

Test results so far have indicated a great potential for the application of diatomaceous biosilica as a component of solar cells, in place of expensive titanium dioxide [37,46,47,48]. An extremely interesting but not yet fully developed idea is the ability to modify the structure of diatomaceous biosilica. There are two basic methods for the functionalization of diatoms [49]. The first one is the in vitro method involving the attachment, via a condensation reaction, of functional groups on the surface of the diatomaceous frustule after its purification, i.e., the removal of the organic matrix of the diatomaceous cell. The second one is the in vivo method based on the stable incorporation of the modifying element into the nanostructural architecture of diatomaceous biosilica during cultivation [50]. The in vitro method can be used to give magnetic properties to diatom frustules by adding iron nanoparticles treated with dopamine [51], as well as to create antibody matrices that can be applied in such techniques as immunoprecipitation [27]. The functionalization of diatoms in vivo is possible when modifying elements are added to the culture medium. This enables the incorporation of the doping element into the structure of the diatom frustules. So far, a few publications report the ability of diatoms to metabolically introduce metal oxides such as titanium or germanium into the structure of silica frustules [3,19,52,53,54,55,56,57,58,59,60,61,62]. There are also results of initial studies on the possibility of metabolic substitution of silicon atoms with nickel, zirconium, tin, zinc, calcium, aluminum, iron, and europium in diatomaceous biosilica [19,63,64,65,66,67,68,69,70,71].

In this review we summarize and discuss the research published to date on the metabolic insertion of chemical elements with specific functional activity (metals or semimetals) into the diatomaceous biosilica structure. Attention is drawn to the culture conditions (culture medium, type of salt and concentration range of admixed elements, pH), physicochemical properties of the biosilica obtained, the amount embedded and distribution of the element in the biosilica structure, and prospects for the use of the doped biosilica. We hope this work will encourage interest in metabolic insertion as a novel and innovative approach to the synthesis of new materials, where the synthesis itself is “outsourced” to the microorganisms as “microtechnologists” who need only the appropriate conditions and reagents.

## 2. Metabolic Insertion of Diatomaceous Biosilica with Titanium and Germanium Ions

### 2.1. Metabolic Insertion of Diatomaceous Biosilica with Titanium Ions

There is an outstanding interest in bioinspired approaches for the synthesis of semiconductors and metal oxide, especially titanium dioxide nanomaterials, as they offer the opportunity for self-assembly into three-dimensional, hierarchical structures. Cell culture systems have especially been identified as a platform for the biosynthesis of photonic nanostructures [55].

A method for the metabolic insertion of titanium ions into diatom cells, whose scheme is shown in Figure 3 [72], was first developed by C. Jeffryes et al. [55] using an unnamed species representing the genus *Pinnularia*.

The doping process was carried out in two stages in a specially prepared photobioreactor. In the first stage, diatoms were grown without the presence of a titanium precursor in a culture medium until all silicon was taken up (the initial concentration of silicon was 0.5 mM). In the second stage, the culture medium was enriched with a solution containing 30 mM of sodium metasilicate and 0.5–4.5 mM of the soluble titanium compound TiCl_4_, resulting from the dissolution of TiOSO_4_ and NaOH in 500 mM HCl. Skolem [58] followed the same pathway, using a two-stage process to dope the siliceous diatom frustule of *Pinnularia* sp. and *Coscinodiscus* sp. with titanium ions in a photobioreactor. A series of experiments was conducted: The first phase involved testing the different combinations of levels of silicon starvation, and the second stage consisted of adding a solution containing 3.6–8.9 mM Si and 0.36–0.62 mM Ti in the form of TiCl_4_ to the medium. Chauton et al. [57] also used a two-stage process of titanium ion doping on *Pinnularia* sp., and using the same titanium precursor, they initiated the titanium uptake when the silicon concentration in the culture medium decreased to less than 0.5 µm. In the study by Eynde et al. [56], the two-stage scheme of the process of doping *Pinnularia* sp. was analogous, differing only in the timing of the addition of the titanium precursor, which took place at the end of cell growth instead of the time of silicon starvation. A study on the two-stage doping of *Fistulifera solaris* by Maeda et al. [59] used titanium(IV) bis(ammonium lactate)dihydroxide (TiBALDH) as the precursor.

A one-stage doping process has been used by other research groups. In Basharina’s work [19], the culture of *Synedra acus* was carried out in microincubators in which 10 mM Na_2_SiO_3_ and 10 mM TiCl_4_ were added simultaneously to a base solution. A similar approach was used by Lang et al. by adding 0.2–2.0 mM TiBALDH to the culture medium of *Thalassiosira weissflogii*. A comparison of the methods used, the culture parameters, and types of titanium ion precursors is shown in Table 1.

Works describing the effect of introducing titanium into the structure of diatomaceous biosilica [57,58,60] have indicated a lack of toxic ion effect on diatom cells. There was also no evidence of titanium ions interfering with the cell cycle of doped diatoms, and SEM and TEM studies conducted on doped frustules showed properly developed structures without any aberration in the pore system. Only in Basharina’s [19] work can we find information concerning a decrease in mechanical strength of doped biosilica. In most cases, a significant increase in biomass is seen as a result of the metabolic insertion of titanium ions. However, in the experiment conducted by Skolem [58] the yield of diatom biomass was lower when compared with the blank. Research conducted by Eynde et al. [56] on *Pinnularia* sp. cultures showed that inhibition of the cell growth process depends on the type of titanium precursor used in the breeding medium (Figure 4).

Maeda et al. [59] noted that the effect of the titanium precursor on diatom cell growth differs with diatom species. When using TiBALDH as a precursor, the growth of *Phaeodactylum tricornutum* and *Thalassiosira pseudonana* was completely inhibited at 2.0 mM TiBALDH, while the inhibition of *F. solaris* growth at the same concentration of TiBALDH was insignificant. Statistically significant inhibition of *F. solaris* cells growth occurred at 5.0 mM TiBALDH, while in *P. tricornutum* and *T. pseudonana*, this occurred at 1.0 mM and 0.5 mM TiBALDH, respectively. According to Lang’s research [60], the growth of *T. weissflogii* cells was inhibited by 2.0 mM TiBLADH.

Comparing the results of the studies on the incorporation of titanium into the diatom frustules, it can be seen that in each experiment there was an uneven distribution of titanium in the biosilica structure. It has been observed that a higher concentration of titanium is found near the pores than near the rib of the frustule. In addition, the amount of titanium incorporated into doped diatom frustules varies significantly between studies, even when using the same titanium ion precursor. In terms of the atomic percentage, Ti:Si, Jeffryes et al. [51] achieved the largest incorporation of 0.6%, but when considering the concentration of titanium incorporated into diatom frustules (mM Ti), Maeda [59], Van Eynde [56], and Lang [60] all obtained higher values. The highest incorporated concentrations of titanium have been achieved using TiBALDH as a precursor. The results of titanium ion doping of diatomaceous biosilica are presented in Table 2.

### 2.2. Metabolic Insertion of Diatomaceous Biosilica with Germanium Ions

There is a notable interest in imbedding nanoscale germanium into dielectric silica for optoelectronic applications. The controlled metabolic insertion of germanium into the silica frustule may produce a silicon/germanium nanocomposite imbedded into the exoskeleton microstructure. This Si–Ge nanocomposite could impart optoelectronic properties to this three-dimensional structure and at the same time controllably alter the microconstruction [62]. Early research into the germanium content in diatoms was focused on the toxicity of this element to diatom cells, and in particular its inhibitory effect on diatom frustule formation [73,74,75]. Lewin [73] noted that a content of only 1.0 µM GeO_2_ significantly inhibited diatom growth, but the diatom species least sensitive to the inhibitory influence of GeO_2_ was *P. tricornutum*, the least silicified of the diatoms studied. It turned out that the inhibitory effect of GeO_2_ on diatom growth can be reduced by adding a correspondingly larger amount of SiO_2_ to the culture medium [73]. This conclusion is also in line with the assumption made by Richter [76] that diatoms show absolute demand for SiO_2_ in their growth phase. These results, and the chemical similarity of germanium and silicon, may suggest that the toxic effects of germanium involve inhibition of the formation of siliceous frustules of diatoms. The influence of germanium on metabolic processes of diatoms was demonstrated by Werner [74], who indicated that Ge(OH)_4_ completely inhibits the synthesis of chlorophyll in *Cyclotella crypitica* and, to a lesser extent, the synthesis of proteins. Similar conclusions were drawn by Azam [75], who showed that high concentrations of Ge(OH)_4_ inhibited the synthesis of chlorophyll and the photosynthetic carbon fixation by diatoms. Basharina et al. [19] also confirmed the toxic effects of germanium ions on diatom cells. The inclusion of germanium in the structure of diatom frustules resulted in various degrees of irregularity; the shape and thickness of frustules was altered, and something resembling an additional layer of silica could be detected. Mubarak Ali et al. [54] also demonstrated a positive relationship between the concentration of germanium in the culture medium of *Stauroneis* sp. strain and the degree of frustule aberration. Qin [52] reported that the metabolic insertion of germanium reduced the pore diameter in diatom frustules. However, Gale et al. [53] showed that metabolic doping of germanium resulted in smaller pores merging into larger ones, taking on the form of fissures. According to Basharina [19], the toxic effect of Ge(OH)_4_ may be associated with the premature condensation of Si(OH)_4_, which occurs without cellular control and causes solid silica deposits to be formed in the wrong places.

In the studies conducted on the metabolic insertion of germanium ions into diatomaceous biosilica, both two-stage [52,53,54,61,62] and one-stage diatom culture methods were used [3,19]. A summary of the diatom culture conditions and the degree of incorporation of germanium is presented in Table 3.

In all works concerning the metabolic incorporation of germanium, its uneven distribution in the diatomaceous biosilica structure was detected [3,19,52,53,54,61,62]. Jeffryes [62] noticed that germanium was dispersed in silica in the form of clusters on both submicrometer and nanometer scales. Similar results were obtained by Mubarak Ali [54], indicating a lack of homogeneity in the distribution of germanium in diatom cells. The germanium content of the structure of the diatom frustule is around 1.0% by weight in all previously published works, which also show the dependence of the amount of germanium in the silica frustule on the initial Ge:Si concentration ratio in the culture medium (Figure 5).

Exceeding a certain limit of the Ge:Si concentration ratio (Ge:Si = 1:100) causes a decrease in the incorporation of germanium into the frustules’ structure [3]. A reduction in the degree of incorporation of germanium ions into the frustule is also visible when the Ge:Si ratio is lower than Ge:Si = 1:4.5 [61]. The most optimal Ge:Si ratio, which shows a linear relationship between the amount of germanium incorporated and the amount of biomass, is Ge:Si = 0.1 [75].

## 3. Metabolic Insertion of Other Metals and Semimetals Ions

The works described above suggest that the concept of using the unique ability of diatom cultures to take up soluble metals and incorporate them into the structure of their frustules can also be applied to other metals, including nickel, europium, aluminum, zinc, iron, calcium, zirconium, and tin [19,63,64,65,66,67,69,70,71].

A summary of the diatom culture conditions used and the degree of incorporation of the indicated elements into diatomaceous biosilica in vivo is presented in Table 4 (calcium) and Table 5 (other elements).

### 3.1. Doping of Biosilica with Aluminium Ions

Machill et al. [67] conducted research on the introduction of aluminum into the frustule structure of the marine diatom, *Stephanopyxis turris*, using artificial seawater containing different concentrations of aluminum (10.5, 42.5, 105.5, and 1055 µm) in the form of AlCl_3_. These concentrations correspond to Al:Si mass ratios of 1:10, 1:2.5, 1:1, and 10:1 respectively. It was observed that a concentration of 10.5 µm avoids uncontrolled aluminum precipitation. A SEM analysis performed to assess the influence of aluminum on the morphology of diatom frustules did not show any significant morphological differences compared with diatoms grown without aluminum in the culture medium. The same size and structure of frustules were detected for both alumina-enriched and natural diatom samples. The incorporation of aluminum into biosilica was detected by ICP-OES analysis of extracted frustules. Quantification has shown that the amount of aluminum embedded in frustules increased rapidly in a medium enriched with aluminum, although no value for the content of aluminum is given. It was observed that the greatest ratio of Al:Si obtained was 1:15. Diatomaceous biosilica doped with aluminum ions can be very valuable material used in catalysis due to its high catalytic activity [51].

### 3.2. Doping of Biosilica with Nickel Ions

Townley et al. [63] presented the results of studies on doping *Coscinodiscus wailesii* cells with nickel ions to modify the optical properties of their frustules. Selected diatom species were cultured in sterile filtered seawater containing an Alga-Gro medium with nickel sulphate added at 5.0, 1.0, 0.5, and 0.1 mg/L. It was observed that the maximum concentration of nickel, which had no significant effect on the growth of diatoms, was 0.5 mg/L.

The SEM analysis of diatoms enriched in nickel showed that the pores in their frustules were more irregular, larger, and less uniform in shape. Studies of the cytoplasmic morphology of *C. wailesii* cultivated in the presence of nickel sulphate showed an interruption of thylakoid stacks and swelling of mitochondria. As the nickel concentration in the culture medium increased, the photoluminescence of silica frustules was also extinguished. The nickel content in the diatoms frustules was about 0.1% by weight and was confirmed by the EDX method. Diatomaceous biosilica doped with nickel ions can be used in biotechnology applications due to its unique optical properties.

### 3.3. Biosilica Doping with Europium Ions

Silicate-based phosphors are promising luminescent materials because of their chemical stability, moisture resistance, and low cost. These materials can be used in various display technologies, such as fluorescent lamps, plasma display panels, field emission displays, and cathode-ray tubes. Zhang et al. [71] published results on the doping of diatomaceous biosilica with europium. Doping was carried out by culturing *Navicula* sp. with the addition of europium hexavalent nitrate (Eu (NO_3_)_3_^∙^6H_2_O) in the molar ratio 1:4 Eu:Si. The culture was carried out for 96 h, then the diatom cells were extracted with ethanol to remove the alcohol-soluble organic material, and the solid residues were heat annealed in air at 1000 °C. An XRD analysis showed the presence of europium in the form of Eu_2_O_3_ and Eu_2_SiO_5_. Europium-doped biosilica exhibited photoluminescent properties with red light emission (614 nm) and excitation at 394 nm corresponding to the wavelength of LED emission.

### 3.4. Doping of Biosilica with Calcium Ions

Leone et al. [65] recently published results on the doping of diatomaceous biosilica with calcium cations for biomedical materials. The idea for this work was based on the knowledge that fibroblasts and osteoblasts grow very well on silica or ceramic substrates, and the presence of calcium ions stimulates the growth of the cells. Calcium-doped diatomaceous biosilica was obtained during the cultivation of *Thalassiosira weissflogii* in autoclaved and ultrafiltered sea water at a controlled temperature of 18–22 °C, with the addition of calcium in the form of CaCl_2_ (14 mM) to the culture medium.

SEM showed that the addition of calcium to diatom culture does not affect their shape and structure. FTIR showed that even though the calcium is not covalently bonded with the biosilica obtained, it remains in the frustules despite the action of 30% hydrogen peroxide to remove organic matter from diatom cells. The detected content of calcium in the doped frustules was in the range of 0.9 ± 0.05% weight. Cell viability studies also confirmed that calcium-doped biosilica can serve as an effective substrate for the growth of fibroblasts and osteoblasts with possible applications in regenerative medicine.

Li et al. [66] also investigated the calcium ion doping of diatoms. *Coscinodiscus* sp. were cultured in ultrafiltered and autoclaved sea water enriched with a f/2 Guillard medium at 21 °C, and calcium inclusion in diatom frustules was achieved by introducing CaCl_2_ calcium chloride into the culture medium at 0.125, 0.25, 0.50, 1.0, and 2.0 mM.

The presence of calcium ions in the structure of diatom frustules was confirmed by XRD and EDXS, but no specific values were provided. Similar to Leone [65], the presence of calcium ions in the culture medium in Li et al. [62] did not cause any significant changes in the morphology of diatoms, and the authors indicated that the material obtained could be used as a haemostatic. Detailed data on the conditions of doping of diatomaceous biosilica with calcium ions are given in Table 4.

### 3.5. Doping of Biosilica with Zirconium and Tin Ions

The influence of zirconium and tin on the growth, morphology, and chemical composition of the freshwater diatom *Synedra acus* was studied by Basharina et al. [19]. Microincubators containing a culture medium with 10 mM Na_2_SiO_3_ and 10 mM Na_2_SnO_3_ or ZrCl_4_ were used. It was noted that doping with zirconium and tin ions caused a slight decrease in growth rate, irregularity of frustules, and a decrease in mechanical strength of the frustules. Analysis showed the presence of 3.4% mol Zr/Si and 0.91% mol Sn/Si in the diatom biomass, and 0% Zr and 0.13% Sn in the purified diatomaceous biosilica.

In order to obtain nanoporous composites of diatom-ZrO_2_, Gannavarapu et al. [64] conducted studies on the culture of the species *Phaeodactylum tricornutum* using artificial sea water at pH = 9 with the addition of 0.8 mM ZrOCl_2*_8H_2_O as a culture medium. The presence of ZrO_2_ zirconium oxides on the surface of *Phaeodactylum tricornutum* frustules was confirmed by EDX analysis. A significant decrease in the size of diatom cells was also observed. The obtained composite has been successfully tested as an electrochemical sensor for the detection of methyl parathion—an organophosphorus pesticide.

### 3.6. Biosilica Doping with Zinc and Iron Oons

In order to verify that diatoms were able to incorporate significant amounts of zinc and iron, Ellwood and Hunter [70] grew the marine diatom *Thalassiosira pseudonana* in sea water with the addition of Zn or Fe salts. A positive relationship was observed between the concentration of free Zn^2+^ and the growth rate of *T*. *pseudonana*. It was also found that limiting access to free Zn^2+^ caused a decrease in cell size in *T. pseudonana* in comparison with those in the free Zn^2+^ replete medium. An analysis of the chemical composition of the biosilica obtained confirmed the inclusion of zinc and iron ions into the structure of the diatom frustules. In the case of zinc ion incorporation, it was found that the amount of Zn^2+^ uptake into the diatom was directly related to the amount of zinc absorbed by the diatom diatom ($2-5\times 10^{-17}\frac{mol}{cell\times day}$), which in turn was directly related to the concentration of free Zn^2+^ in the culture medium. The relationship between Zn/Si and free Zn^2+^ suggests that the Zn content of diatom shells may be useful in reconstructing changes in oceanic concentration of free Zn^2+^. In the case of iron ion incorporation, no direct proportional relationship between the incorporated iron content and its concentration in the culture medium was detected. Moreover, no specific value was determined for the amount of iron incorporated into the diatom cell.

The studies reported in Ellwood and Hunter [70] were continued by Jaccard et al. [69]. They cultivated the freshwater diatom *Stephanodiscus hantzschii* (UTCC 267) in a modified medium CHU-10 [77] with the addition of the Zn–EDTA complex, containing 10^−10.6^–10^−7.6^ M zinc ions. The presence of zinc ions in the structure of diatom frustules was confirmed by ICP-MS. No value was given for the amount of zinc incorporated into the diatom frustules, but it was found that the highest degree of zinc ion incorporation was achieved at a concentration of 10^−8.5^ M Zn^2+^.

The conditions used to grow diatoms and the degree of metabolic insertion of elements (Al, Ni, Eu, Zr, Sn, Zn, Fe) into diatomaceous biosilica are summarized in Table 5.

## 4. Conclusions

This analysis of the limited research available on the metabolic insertion of elements into the structured diatomaceous biosilica indicates the emergence of a new trend in synthesis methods for next-generation functional materials. This new approach harnesses the ability of microorganisms to act as microtechnologists for the synthesis of smart materials, in particular the diatoms in synthesizing new silica materials with desirable properties.

There is evidence that many factors, such as the type of precursor, chemical composition and pH of the culture medium, temperature, illumination, and aeration, can significantly impact the process of metabolic insertion into diatomaceous biosilica, the growth and development of diatom cells, and the structure and morphology of the frustules. Therefore, there is scope for optimizing the process of metabolic insertion by manipulating these factors (e.g., by changing the Ge:Si ratio in the culture medium). A further aspect is the variation in the influence of different factors and doping elements on different diatom species, and therefore the choice of diatom species is also crucial. Finally, there is further potential for metabolic insertion techniques by using a genetic modification of diatom cells [20,78] in order to give them specific abilities for selective metabolic insertion of selected elements.

## Figures and Tables

**Figure 1 materials-13-02576-f001:**
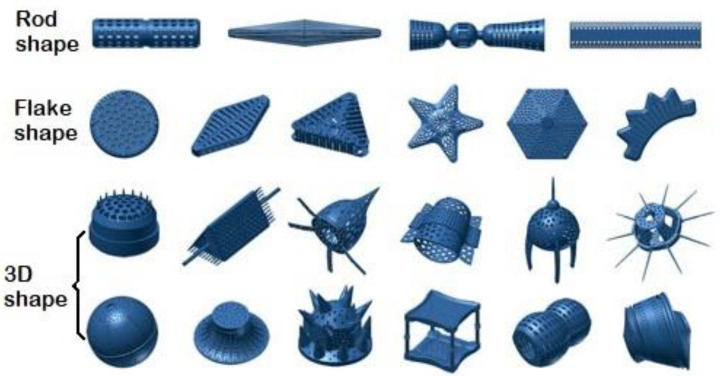
The unique structure of the diatom frustule. The images are 3D models [4].

**Figure 2 materials-13-02576-f002:**
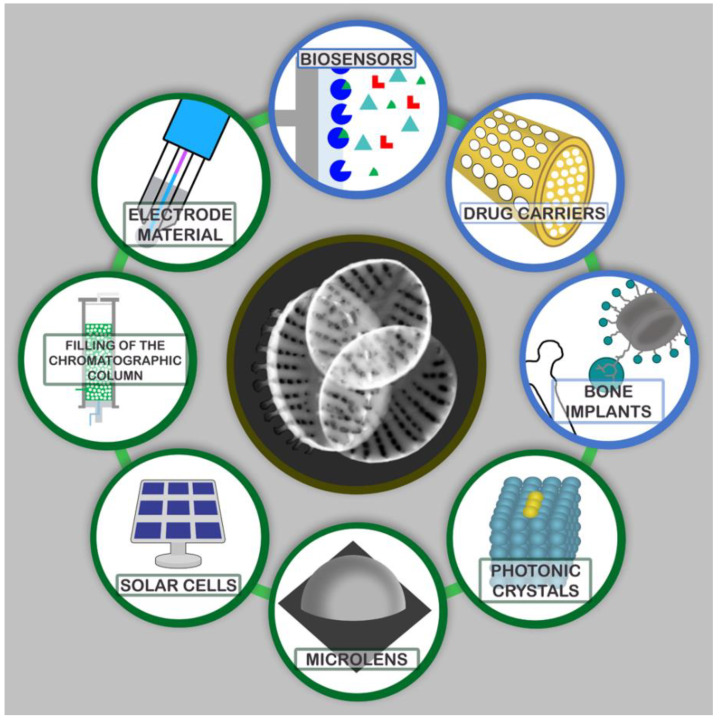
A range of possibilities to use diatomaceous biosilica.

**Figure 3 materials-13-02576-f003:**
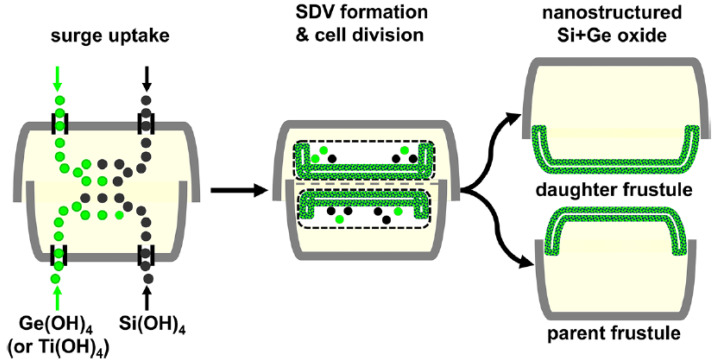
Alleged scheme of metabolic insertion of Ge or Ti into the diatom frustule during cultivation.

**Figure 4 materials-13-02576-f004:**
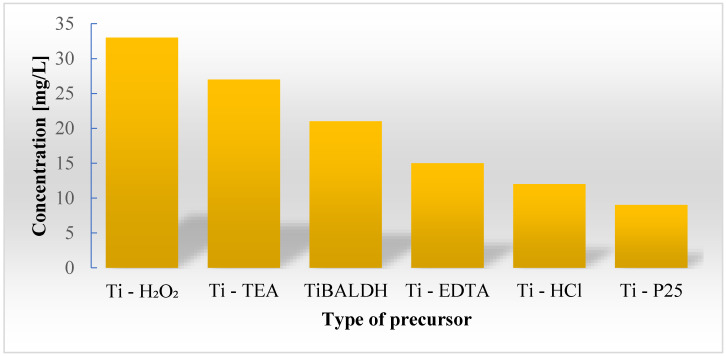
The limit of inhibition concentration depending on the type of titanium precursor used: Ti–H_2_O_2_: Ti–hydrogen peroxide; Ti–TEA: Ti–triethanolamine; TiBALDH: titanium(IV) bis(ammonium lactate)dihydroxide; Ti–EDTA: Ti–ethylenediaminetetraacetic acid; Ti–HCl: acid digested hydrolyzed titania; Ti–P25: titanium(IV) oxide.

**Figure 5 materials-13-02576-f005:**
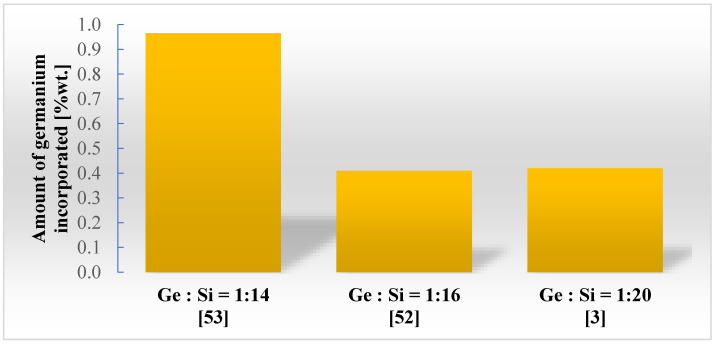
The dependence of the amount of germanium to silica frustule on the initial Ge:Si concentration ratio in the culture medium [3,52,53].

**Table 1 materials-13-02576-t001:** Summary of methods used for the metabolic insertion of titanium ions to diatomaceous biosilica.

Ref. ^a^	[19]	[55]	[56]	[57]	[58]	[59]	[60]
Species	*S.ac.* ^b^	*P.* sp. ^c^	*P.* sp.	*P.* sp.	*P.* sp.; *Cos.* sp. ^d^	*F. sol.* ^e^	*T. weiss.* ^f^
Culture Medium	DM	f/2	WC	f/2	f/2	f/2	f/2
Lux [μmol/m^2^_∙_s]	13–16	149	30	130	130	140	246
T [°C]	12	22	20	20	20	25	16–22
pH	7.4	8.4–8.6	7.6–8.4	8.0–8.4	8.0–8.35	6.4	No data
Process type	I ^g^	II ^h^	II	II	II	II	I
Precursor	TiCl_4_	TiOSO_4_ ^i^	TiBALDH	TiOSO_4_	TiOSO_4_	TiBALDH	TiBALDH
[Si] [mM] ^j^	10	0.48	No data	6.2	3.6–8.9	No data	0.2
[Ti] [mM] ^k^	10	0.0085–0.073	0.0–0.56	0.36	0.36–0.62	0.25–20	0.2–2.0

Notes: ^a^ References, ^b^
*Synedra acus*, ^c^
*Pinnularia* sp., ^d^
*Coscinodiscus* sp., ^e^
*Fistulifera solaris*, ^f^
*Thalassiosira weissflogii*, ^g^ One-stage process, ^h^ Two-stage process, ^i^ TiOSO_4_+NaOH/HCl, ^j^ Si content in the culture medium, ^k^ Ti content in the culture medium.

**Table 2 materials-13-02576-t002:** Summary of applied conditions for the cultivation of diatoms and the degree of incorporation of titanium into diatomaceous biosilica.

Ref.	[19]	[55]	[56]	[57]	[58]	[59]	[60]
Species	*S. ac.*	*P.* sp.	*P.* sp.	*P.* sp.	*P.* sp. *Cos.* sp.	*F. sol.*	*T. weiss.*
Precursor	TiCl_4_	TiOSO_4_	TiBALDH	TiOSO_4_	TiOSO_4_	TiBALDH	TiBALDH
[Ti] [mM]	10	0.0085–0.073	0.0–0.56	0.36	0.36–0.62	0.25–20	0.2–2.0
Ti:Si [% atom] ^a^	0.16	0.6	3.2	0.62	0.34	6.02	20
Ti:Si [%wt] ^b^	0.6	2.3	10.4	2.37	0.93	10.6	34
Method of Analysis	ICP-MS	ICP-AES	ICP-MS	ICP-MS	EDS	ICP-AES	EDS

Notes: ^a^ percentage by atomic of biosilica, ^b^ percentage by weight of biosilica.

**Table 3 materials-13-02576-t003:** Comparison of the applied conditions for diatoms culture and the degree of germanium incorporation into diatomaceous biosilica.

Ref.	[3]	[19]	[52]	[53]	[54]	[61]	[62]
Species	*T. pseudo.* ^a^	*S. ac.*	*N. frust.*	*P.* sp.	*Stauro.* sp. ^b^	*N. frust.* ^c^	*P.* sp.
Culture Medium	No data	DM	LDM	LDM	f/2	LDM	LDM
Lux [μmol/m^2^_*_s]	No data	13–16	150	150	164	125	50
T [°C]	No data	12	22	22	No data	22	22
pH	No data	7.4	8.4–8.9	No data	No data	8.2–8.4	8.3
Process Type	I	I	II	II	II	II	II
Precursor	Ge (OH)_4_	Na_2_GeO_3_	GeO_2_	GeO_2_	GeO_2_	Ge (OH)_4_	GeO_2_
[Ge] ^d^ [mM]	0.1	0.11	0.72	0.78	1.07	0.384	0.53
Ge_incorp._ ^e^	0.42% wt.	5.1%mol Ge/Si	0.411%wt.	0.965%wt.	No data	2.74 mg Ge/g of DCW ^f^	0.965%wt.
Method of Analysis	ICP-OES	ICP-MS	ICP	ICP	EDS	EDS	ICP

Notes: ^a^
*Thalassiosira pseudonana*, ^b^
*Stauroneis* sp., ^c^
*Nitzschia frustulum*, ^d^ germanium content in the culture medium, ^e^ content of germanium incorporated into the diatom frustules, ^f^ dry cell weight.

**Table 4 materials-13-02576-t004:** Summary of parameters for doping of diatomaceous biosilica with calcium ions.

Ref.	[65]	[66]
Species	*T. weiss.*	*Cos.* sp.
Precursor	CaCl_2_	CaCl_2_
Culture Medium	f/2	f/2
Lux [μmol/m^2^_*_s]	No data	246
T [°C]	18–22	21
pH	No data	No data
Ca_incorp_ ^a^ [%wt]	0.9	No data
Method of Analysis	EDX	EDXS

Note: ^a^ content of calcium incorporated into the diatom frustules.

**Table 5 materials-13-02576-t005:** Summary of results of doping diatoms with metal ions.

	Zr/Sn		Ni	Al	Zn		Fe	Eu
Ref.	[19]	[64]	[63]	[67]	[69]	[70]	[70]	[71]
Species	*S. ac.*	*P. trico.* ^a^	*Cos.*	*S. tur.* ^b^	*S. hanz.* ^c^	*T. pseudo.*	*T. pseudo.*	*Navi.* sp. ^d^
Culture Medium	DM	Aquil	Alga-Gro	ASW	CHU-10	f/2	f/2	“Ningbo 3”
Lux [μmol/m^2^_*_s]	13–16	No data	No data	82	50	120	120	246
T [°C]	12	19	22	18	20	20–22	20–22	25
pH	7.4	9	No data	8.0–8.2	6.4	7.2–8.3	8.0	No data
Process Type	I	I	I	I	II	II	II	II
Precursor	ZrCl_4_	ZrOCl_2*_8H_2_O	NiSO_4_	AlCl_3_	Zn–EDTA	Zn–EDTA	Fe–EDTA	Eu (NO_3_)_3_^∙^6H_2_O
X_incorp._ **^e^**	3.4%mol Zr/Si; 0.91%mol Sn/Si	No data	~0.1%wag	No data	No data	2–5 ^.^ 10^17^ mol Zn/cell·day	No data	No data
Method of Analysis	ICP-MS	EDX	EDX	ICP-OES	ICP-MS	GFAAS	GFAAS	XRD

Notes: ^a^
*Phaeodactylum tricornutum*; ^b^
*Stephanopyxis turris*; ^c^
*Stephanodiscus hantzschii*; ^d^
*Navicula* sp.; ^e^ the content of the element incorporated into the diatom frustule, where X = Al, Ni, Eu, Zr, Sn, Zn, Fe.

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
