# Peer review of "“Outsourcing” Diatoms in Fabrication of Metal-Doped 3D Biosilica"

_materials, 2020, doi:10.3390/ma13112576_

Round 1

Reviewer 1 Report

The manuscripts "“Outsourcing” diatoms in fabrication of metal-doped 3D biosilica" report a literature study regarding the metal doping of 3D biosilica produced by various microorganisms. It is an interesting study and can bring to the readers the literature overview on the one-step synthesis of doped biosilica directly in bio-sources. Before to recommend the publication, some comments are addressed:

  • In the Table 3, 3rd column, line 8: the authors should revise the precursor TiCl4, as this table provide information about "Comparison of the applied conditions for diatoms culture and the degree of germanium incorporation into diatom biosilica".
  • Also, regarding "Table 4. Summary of parameters for doping of diatom biosilica with calcium ions.", a correction should be made to the footnote "a content of germanium incorporated into the diatom frustules."
  • In the whole manuscript the therm "in vivo" has to be written in italic as "in vivo"
  • As one of the keywords is "Metabolic Inserting", a recommendation is to insert in the manuscript a suggestive scheme of metabolic reactions where the metals are inserted in the biosilica structure.
  • What was the reason of separation the metal ions in two section?: "2. Metabolic insertion of diatom biosilica with titanium and germanium ions" and "4. Metabolic insertion of other metals and semi-metals ions".
  • It seems there is missing one section: 3.
  • Did the authors try to classify somehow the used metal in biosilica dopping?

Author Response

Response to Reviewer #1:

Thank very much you for your willingness to review our manuscript and valuable comments.

Remark 1: “In the Table 3, 3rd column, line 8: the authors should revise the precursor TiCl4, as this table provide information about "Comparison of the applied conditions for diatoms culture and the degree of germanium incorporation into diatom biosilica".

Answer: Thank you for pointing out the incorrectly used precursor. This error has been corrected.

Remark 2: Also, regarding "Table 4. Summary of parameters for doping of diatom biosilica with calcium ions.", a correction should be made to the footnote "a content of germanium incorporated into the diatom frustules."

Answer: Thank you for noting the incorrect footnote in Table 4. This error has been corrected.

Remark 3: In the whole manuscript the therm "in vivo" has to be written in italic as "in vivo".

Answer: Thank you for paying attention to this aspect. Throughout the manuscript, the in vivo and in vitro terms have been corrected and printed in italics.

Remark 4: As one of the keywords is "Metabolic Inserting", a recommendation is to insert in the manuscript a suggestive scheme of metabolic reactions where the metals are inserted in the biosilica structure.

Answer: Thank you for paying attention to the aspect of the graphical representation of the metabolic insertion process. A putative scheme for the metabolic insertion of germanium or titanium has been inserted into the manuscript and appears there as Figure 3.

Remark 5: What was the reason of separation the metal ions in two section?: "2. Metabolic insertion of diatom biosilica with titanium and germanium ions" and "4. Metabolic insertion of other metals and semi-metals ions".

Answer: Separation of metal ions in two sections was caused by the fact that the amount of works related to the metabolic insertion of titanium and germanium into the diatoms frustules is much greater than the works related to doping diatoms with other metal ions. By dividing the description of doping with titanium and germanium and other metals into two sections, we wanted to emphasize the above fact.

Remark 6: It seems there is missing one section: 3.

Answer: Thank you for noting the incorrect numbering of the main sections of the manuscript. This error has already been corrected.

Remark 7: Did the authors try to classify somehow the used metal in biosilica dopping?

Answer: No, we have not attempted to classify the metals used in biosilica doping. In the discussed works doped metals belong to too many different groups in terms of chemical properties that it would be difficult to find a common feature for these elements (apart from doping), against which we could make a classification.

Reviewer 2 Report

In this manuscript, the authors carry out an exhaustive review based on the metabolic insertion of chemical elements into diatom biosilica, paying special interest to the experimental conditions (pH, culture medium, precursor, etc.) used in each case.

In my opinion the content is interesting for the scientific community, however, there are several minor and major revisions that must be done before being accepted.

  1. The manuscript is about biosilica, however, there is no doubt the similarity with mesoporous silica nanomaterials. The incorporation of a brief description of said materials (10.3390/pharmaceutics10040279) as well as the similarities / differences and advantages / disadvantages of both materials (10.1016/j.jconrel.2018.05.013) would bring great added value to the article.

  1. Two basic approaches exist to functionalize diatoms (10.3390/molecules22122232): (i) in vivo by genetic engineering or by changing growth parameters during the cultivation; (ii) in vitro by chemical or physical treatment after cell death. The authors focus on the first one, however, it is important to mention both methods as well as the advantages and disadvantages of each of them.

  1. At the beginning of section 4.4 Doping of biosilica with calcium ions, the authors briefly explain the utility of incorporating calcium into the structure. This wisely structure should be included for each chemical element introduced, especially for Ti and Ge. In other sections the authors refer to some property of the new material “Europium-doped biosilica exhibited photoluminescent properties with red light emission (614 nm) and excitation at 394 nm”, but the purpose is not clear.

  1. The method used to calculate the amount of metal incorporated in the biosilica is very important, therefore, as done in Table 2, it should be added to the rest of the tables.

  1. It is not clear why the authors have incorporated references in the culture medium row of Table 1. Was it an error? If not, please explain why.

In addition, there are some minor type errors that should be corrected:

  • Some typo errors are present:
  1. Duplicated “in” (line 188)
  2. abberation instead of aberration. (line 145)
  3. f %at. (a gap is needed) (line 172)

I recommend this manuscript to be accepted after major revision.

Author Response

Response to Reviewer #2:

Thank very much you for your willingness to review our manuscript and valuable comments.

Remark 1: The manuscript is about biosilica, however, there is no doubt the similarity with mesoporous silica nanomaterials. The incorporation of a brief description of said materials (10.3390/pharmaceutics10040279) as well as the similarities / differences and advantages / disadvantages of both materials (10.1016/j.jconrel.2018.05.013) would bring great added value to the article.

Answer: Thank you for the advice on how we can increase the scientific value of our article. We have included a brief description of mesoporous silicate materials, including their comparison with diatomaceous biosilica and paying attention to their advantages and disadvantages.

„Currently, diatomaceous biosilica, due to its three-dimensional, porous structure, wide availability and the possibility of biosynthesis through the cultivation of diatoms under artificial conditions, is one of the most frequently used substitutes for mesoporous silica materials in modern technologies. These materials, despite their biocompatibility and large specific surface area [22] are difficult to synthesize because of necessity of considerable financial input, a large amount of energy and association of toxic materials using [23]”.

Remark 2: Two basic approaches exist to functionalize diatoms (10.3390/molecules22122232): (i) in vivo by genetic engineering or by changing growth parameters during the cultivation; (ii) in vitro by chemical or physical treatment after cell death. The authors focus on the first one, however, it is important to mention both methods as well as the advantages and disadvantages of each of them.

Answer:  Thank you for paying attention to mention also the in vitro method as part of the in vivo comparison. We have included a brief description of the both methods, including the possibility of their application.

„Extremely interesting, but not yet fully developed, is the ability to modify the structure of diatomaceous biosilica. There are two basic methods for the functionalization of diatoms [49]. The first one is the in vitro method involving the attachment, via a condensation reaction, of functional groups on the surface of the diatomaceous frustule after its purification, i.e. removal of the organic matrix of the diatomaceous cell. The second one is the in vivo method based on the stable incorporation of the modifying element into the nanostructural architecture of diatomaceous biosilica during cultivation [50]. The in vitro method can be used to give magnetic properties to diatom frustules by adding iron nanoparticles treated with dopamine [51], as well as to create antibody matrices that can be applied in such techniques as immunoprecipitation [27]. Functionalization of diatoms in vivo is possible when modifying elements are added to the culture medium. This enables the incorporation of the doping element into the structure of the diatom frustules.

Remark 3: At the beginning of section 4.4 Doping of biosilica with calcium ions, the authors briefly explain the utility of incorporating calcium into the structure. This wisely structure should be included for each chemical element introduced, especially for Ti and Ge. In other sections the authors refer to some property of the new material “Europium-doped biosilica exhibited photoluminescent properties with red light emission (614 nm) and excitation at 394 nm”, but the purpose is not clear.

Answer: Thank you for paying attention to attach to each introduced chemical element a description of the suitability of including it in the structure of diatomaceous silica. To each section describing the introduction of a given chemical element to the structure of diatomaceous silica, a brief description of the usefulness of incorporating it into the structure has been added.

Titanium: „There is outstanding interest in bioinspired approaches for synthesis of semiconductor and metal oxide, especially titanium dioxide nanomaterials, as they offer the opportunity for self-assembly into three-dimensional, hierarchical structures. Especially, cell culture systems have been identified as a platform for the biosynthesis of photonic nanostructures [52].”

Germanium: „There is notable interest in imbedding nanoscale germanium into dielectric silica for optoelectronic applications. The controlled metabolic insertion of germanium into the silica frustule may produce a silicon/germanium nanocomposite imbedded into the exoskeleton microstructure. This Si–Ge nanocomposite could impart optoelectronic properties to this three-dimensional structure and at the same time controllably alter microconstruction [59].”

Aluminium: " Diatomaceous biosilica doped with aluminum ions can be a very valuable material used in catalysis due to its high catalytic activity. [51]”

Nickel: „Diatomaceous biosilica doped with nickel ions, due to its unique optical properties can be used in biotechnology applications.”

Europium: „Silicate-based phosphors are promising luminescent materials because of their chemical stability, moisture resistance and low material cost. These materials can be used in various display technologies such as: fluorescent lamps, plasma display panels, field emission displays and cathode-ray tubes.”

„Europium-doped biosilica exhibited photoluminescent properties with red light emission (614 nm) and excitation at 394 nm, which corresponds to the wavelenght of LED emission.”

Zinc: „The relationship between Zn:Si and free Zn2+ suggests that the Zn content of diatom shells may be useful in reconstructing changes in oceanic concentration of free Zn2+.”

Remark 4: The method used to calculate the amount of metal incorporated in the biosilica is very important, therefore, as done in Table 2, it should be added to the rest of the tables.

Answer: Thank you for paying attention to the addition to each table methods used to calculate the amount of metal contained in the structure of diatomaceous biosilica. We have added an additional row to each table containing data on the methods used to calculate the amount of metal contained in the diatomaceous silica structure.

Remark 5: It is not clear why the authors have incorporated references in the culture medium row of Table 1. Was it an error? If not, please explain why.

Answer: Thank you for paying attention to the references left in the row concerning the information about culture medium. We apologize for this mistake - it appeared in the draft version, although in the final version it has already been removed.

minor errors:

Remark: Some typo errors are present:

  • Duplicated “in” (line 188)
  • abberation instead of aberration. (line 145)
  • f %at. (a gap is needed) (line 172)

Answer: Thank you for pointing out the minor errors found in the entire manuscript. All of them have been corrected.

Round 2

Reviewer 1 Report

Now, the manuscript can be considered for publication in Materials.

Reviewer 2 Report

After the corrections made by the authors, the quality of the manuscript has been improved and I consider that it will be of great interest to the scientific community. I recommend to accept it in the present form.